# Extracellular Matrix Biomimetic Hydrogels, Encapsulated with Stromal Cell-Derived Factor 1, Improve the Composition of Foetal Tissue Grafts in a Rodent Model of Parkinson’s Disease

**DOI:** 10.3390/ijms23094646

**Published:** 2022-04-22

**Authors:** Vanessa Penna, Niamh Moriarty, Yi Wang, Kevin C. L. Law, Carlos W. Gantner, Richard J. Williams, David R. Nisbet, Clare L. Parish

**Affiliations:** 1The Florey Institute of Neuroscience and Mental Health, The University of Melbourne, Parkville, VIC 3052, Australia; vanessa.penn@gmail.com (V.P.); niamh.moriarty@unimelb.edu.au (N.M.); kevin.law@florey.edu.au (K.C.L.L.); cg731@cam.ac.uk (C.W.G.); 2The Graeme Clark Institute, The University of Melbourne, Parkville, VIC 3052, Australia; yi.wang24@unimelb.edu.au (Y.W.); david.nisbet@unimelb.edu.au (D.R.N.); 3Department of Biomedical Engineering, Faculty of Engineering and Information Technology, The University of Melbourne, Parkville, VIC 3052, Australia; 4Research School of Engineering, The Australian National University, Canberra, ACT 2601, Australia; 5iMPACT, School of Medicine, Deakin University, Waurn Ponds, VIC 3216, Australia; richard.williams@deakin.edu.au

**Keywords:** stem cells, transplantation, dopamine, Parkinson’s disease, biomaterials, self-assembling peptide, hydrogel, laminin, stromal cell-derived factor 1, SDF1

## Abstract

Clinical studies have provided evidence for dopamine (DA) cell replacement therapy in Parkinson’s Disease. However, grafts derived from foetal tissue or pluripotent stem cells (PSCs) remain heterogeneous, with a high proportion of non-dopaminergic cells, and display subthreshold reinnervation of target tissues, thereby highlighting the need to identify new strategies to improve graft outcomes. In recent work, Stromal Cell-Derived Factor-1 (SDF1), secreted from meninges, has been shown to exert many roles during ventral midbrain DA development and DA-directed differentiation of PSCs. Related, co-implantation of meningeal cells has been shown to improve neural graft outcomes, however, no direct evidence for the role of SDF1 in neural grafting has been shown. Due to the rapid degradation of SDF1 protein, here, we utilised a hydrogel to entrap the protein and sustain its delivery at the transplant site to assess the impact on DA progenitor differentiation, survival and plasticity. Hydrogels were fabricated from self-assembling peptides (SAP), presenting an epitope for laminin, the brain’s main extracellular matrix protein, thereby providing cell adhesive support for the grafts and additional laminin–integrin signalling to influence cell fate. We show that SDF1 functionalised SAP hydrogels resulted in larger grafts, containing more DA neurons, increased A9 DA specification (the subpopulation of DA neurons responsible for motor function) and enhanced innervation. These findings demonstrate the capacity for functionalised, tissue-specific hydrogels to improve the composition of grafts targeted for neural repair.

## 1. Introduction

Parkinson’s disease (PD) is a progressive neurodegenerative disorder where the loss of ventral midbrain (VM) dopamine (DA) neurons underpins disturbances in motor function [1]. While DA pharmacotherapy is the mainstay in treatment, it presents waning efficacy over time and complications associated with systematic delivery. In contrast, cell replacement therapy offers sustained and targeted DA replacement. Preclinical and clinical studies have provided evidence that transplanted DA progenitors, isolated from the developing VM, are capable of surviving, releasing DA and reversing motor deficits [2]. More recently human pluripotent stem cells (hPSCs), differentiated into VM progenitors, have been shown to similarly alleviate motor symptoms in rodent and non-human primates, with success resulting in the recent initiation of several clinical trials [3,4,5,6].

While progress of this therapeutic approach has been exciting, as with all new technologies, there remains room for improvement—see reviews [7,8]. Within the field it is widely accepted that, despite highly efficient refined differentiation protocols, upon transplantation DA progenitors/neurons show poor survival, resulting in low proportions of DA neurons within the grafts [8,9,10]. This poor survival of DA neurons is, in part, dictated by trauma to the cells during both isolation (from either the foetus or in vitro cultures) and their physical implantation, yet also governed by the exposure of the progenitors to the adult host brain environment that is devoid of the many trophic cues normally present during the establishment of new neural networks in development [11,12]. Whilst DA neuron survival is critical for graft function, more specifically is the relative contribution by the A9 neuronal subpopulation that form the nigrostriatal pathway and are responsible for dopamine-modulation of motor function [13].

The past 30 years has seen extensive efforts made to promote the survival and plasticity of DA progenitor grafts in PD models [14], in addition to understanding how to enrich for A9 neurons [15]. More recently, a series of studies have provided new insight into the role of the chemokine Stromal cell-Derived Factor-1, SDF1 (also known as CXCL12), and the meninges in midbrain DA development to suggest that this protein may also improve graft outcomes. Several studies demonstrated that the co-culture of embryonic stem cells on stromal cell feeder layers, or meningeal cells, promoted neural induction and induced DA differentiation via secreted factors termed stromal derived inducing activity (SDIA) [16,17,18,19]. Genome expression analysis revealed that SDF1 was one of the secreted factors [19]. Subsequently SDF1, secreted from the meninges overlying the VM and acting via the CXCR4 receptor expressed on DA cell, was shown to modulate DA progenitor migration and neuritogenesis [20,21,22]. Finally, meningeal cells, co-transplanted with VM progenitors influenced DA maturation, increased A9 fate acquisition and promoted neurite extension in a mouse model of PD [22]. The evident outstanding piece of the puzzle is whether SDF1 can directly influence DA graft outcomes.

In an effort to sustain SDF1 expression during periods of graft implantation and integration, and to circumvent complications associated with sustained protein infusion or viral transduction, here we incorporated the functional protein into a bioengineered hydrogel—an approach previously adopted by us to control protein delivery over defined temporal periods ranging from days, to weeks and even months [23,24,25,26,27].

Additional to the benefits of controlling protein delivery, biomaterials have the capacity to deliver structural and biochemical biomimetics of the extracellular matrix (ECM) that can be advantageous in regenerative medicine to support both host and transplanted cells [28]. These interfacing materials can provide further benefits, dependent on their mode of synthesis and structure, protecting transplanted cells against shear forces during implantation [29], shielding the graft from the host immune system [30], and/or dampen the host immunological response [31]. Whilst a raft of biomaterials have been engineered and trialled for their regenerative capacity, those synthesised through processes that most closely mimic biology are evidently advantageous. With nature utilising supramolecular interactions to drive self-organisation of biological macromolecules such as peptides and proteins, these same principles can be applied to design synthetic peptides that can self-assemble into larger complex structures, like those normally found in biology [32]. Various hydrogels have been formed from self-assembling peptides, yet have mostly utilized long, highly structured peptides such as RADA16, or peptide amphiphiles that require many steps in their synthesis and purification, resulting in increased costs and complexity. Consequently, there has been increasing interest into the fabrication of minimalist peptide sequences (<10 amino acids) that retain the ability to self-assemble and present bioactive sequences. Here we utilize our previously described Fmoc-based self-assembling peptide (SAP) system [33], to present a functional epitope sequence (IKVAV) of the α1 chain of laminin—the main extracellular matrix protein present in the brain. This epitope has been shown to influence cell adhesion, proliferation, migration, differentiation and neurite plasticity [34], as well as influence graft outcomes in the intact and injured brain [27,35,36,37]. Here we utilize the neural tissue-specific IKVAV SAP hydrogel, shear-loaded with SDF1, to assess the beneficial impact on the survival, differentiation, and integration of grafts in an animal model of PD. The adoption of rodent fetal grafting, over human IPSC-derived grafts, provides a valuable resource to accelerate basic discovery, noting graft survival and integration can be assessed within weeks, compared to months after transplantation—recognising that there will be a need for subsequent translation of successful findings into more human relevant models.

## 2. Results

### 2.1. Laminin-Based IKVAV Hydrogel Fabrication and Sustained SDF1 Delivery

Here we utilize the Fmoc self-assembly to fabricate the Fmoc-DDIKVAV (Figure 1B), laminin epitope-presenting hydrogel for in vivo delivery of cells. The fabrication process involved the N-terminus of the peptide sequence being protected by an aromatic fluorenylmethoxycarbonyl (Fmoc) moiety. The aromatic groups of the Fmoc share electrons to form π-π interactions creating the backbone of the structures, while the individual peptides form hydrogen bonds with each other to create a secondary β-sheet protein structure. FTIR analysis of amide I region showed a characteristic peak for β -sheets with an observable carbamate peak (Figure 1C), and the stable formation of a dominant assembly confirmed by circular dichroism spectroscopy, showing transition peak at 200 nm (Figure 1D). As result of several of these assemblies’ individual nanofibres were formed that were nanometers in diameter and microns in length. These nanofibers then established supramolecular associations to create a highly branched nanofibrous network macroscale hydrogel whose structure was be confirmed by transmission electron microscopy, TEM (Figure 1E). Rheological analysis of the elastic shear moduli of storage (G’, black dots) and loss modulus (G”, white dots) validated characteristic viscoelastic gel properties and demonstrated that the resultant gel had a modulus not dissimilar to the rodent brain (grey box, Figure 1F [38]), thereby presenting a hydrogel suitable for brain repair.

Critical for SDF1 to exert effects on the graft was the requirement for sustained presentation of the protein to the donor cells acutely following implantation and during ongoing integration. In vitro, we demonstrated that recombinant SDF1 was rapidly degraded when added directly to the culture media, with less than 1% of the protein remaining after just 30 min. (Figure 1G). By comparison, shear encapsulation of the protein within the SAP hydrogel prolonged presentation, following an initial burst release. The observed burst release of the protein likely resulted from poorly entrapped SDF1 rapidly leaking from the SAP scaffold before it reformed from its solution to gel state, yet had the benefit of prolonging the acute period of protein presentation over 4 hours (Figure 1H), that was beneficial to the cells during the critically vulnerable phase of in vivo delivery. The majority of the SDF1 protein however interacted through absorption with the peptide fibrils within the scaffolds, resulting in a gradual release from the gel over 5–14 days, and corresponding to periods of DA progenitor maturation and integration. Ongoing release, of a slower rate, continued for at least 28 days (Figure 1I). Note, shear loading of SDF1 into the hydrogel had no impact on the assembly of the peptides or stiffness of the gels (data not shown).

### 2.2. SDF1 Functionalised Scaffolds Have No Impact on the Host Inflammatory Response

At 12 weeks after transplantation of primary VM progenitors (±AP ± SDF1) all animals were killed for histological analysis. Grafts could be identified in all brains by GFP labelling, Figure 2A–D. Five animals were excluded from further analysis due to misplacement of the graft (residing outside the target striatal tissue) or due to the presence of evident backflow of the GFP+ cells along the injection site (resulting in the presence of GFP cells in the overlying cortex and/or on the surface of the brain).

While the host inflammatory response to both implanted VM progenitors and hydrogels has previously been examined by us [24,36,37,39], SDF1 has been shown to act as both pro- and anti-inflammatory molecule, during injury and regeneration [40]. We therefore examined the local host inflammatory response to all grafted conditions. Density of GFAP+ reactive astrocytes and CD11b microglia revealed no difference between any of the graft groups (Appendix A) indicating that neither the gel or the mode of SDF1 delivery impacted on the local inflammation.

### 2.3. SDF1 Functionalised Scaffolds Increases DA Differentiation

Volumetric analysis of the grafts revealed that core graft size in the presence of either the SAP hydrogel (0.35 ± 0.08 mm^3^) or sSDF1 (0.33 ± 0.06 mm^3^) remained unchanged compared to grafts of cells alone (0.24 ± 0.02 mm^3^), indicating that neither the gel or acute delivery of the protein impeded or promoted the survival and/or proliferation of the graft, Figure 2I. In contrast, grafts in the presence of the SDF1-functionalised SAP hydrogel (Cells + SAP-shSDF1) were significantly larger (0.44 ± 0.04 mm^3^), Figure 2I, suggesting that the sustained presence of SDF1 influenced the proliferation and/or survival of the implanted progenitors. Staining against the proliferative protein KI67, and apoptotic protein cleaved-caspase-3, showed no difference across all groups (data not shown), suggesting that analysis at earlier time points after grafting are necessary to discern the survival and/or proliferation effects of the functionalised scaffolds on the VM progenitors.

The proportion of NeuN+ neurons within the graft remained unchanged across all treatment groups (ranging on average from 52–65% of the total cells, Figure 2J), indicating that the majority of the graft acquired a maturing neuronal phenotype that was not influenced by either the presence of the hydrogel or SDF1 protein.

Isolation of donor tissue from TH-GFP reporter mice enabled clear visualisation of graft-derived dopaminergic neurons (GFP+), distinct from residual host dopaminergic midbrain system (GFP-). Quantitative assessment of GFP+ dopaminergic neurons within the grafts revealed that the presence of the laminin-based hydrogel, significantly increased the total yield (Cells: 1872 ± 197, Cells + SAP: 3346 ± 102), Figure 2K. Acute delivery of SDF1 had no significant impact on DA neuron numbers (Cells + sSDF1: 3028 ± 362). In contrast, the presences of the SDF1 functionalised SAP scaffold resulted in a 2.6-fold increase in GFP+ DA neurons (4934 ± 321) compared to Cells alone, and increased compared to Cells + SAP, indicative of the synergistic effect of the SDF protein and the laminin-based scaffold, Figure 2K. Assessment of GFP+ cell density revealed an increase trend for grafts in the presence of the SAP, yet only significantly increased in Cells + SAP-shSDF1 grafts (Figure 2L). Such observations suggest that the increased yield of TH-GFP+ neurons within these grafts was over and above the increase in graft volume, and that the functionalised scaffold was promoting DA differentiation.

### 2.4. Functionalised Scaffolds Have No Impact on the Density of Dopaminergic Innervation of the Host Striatum

Additional to the benefit of quantifying DA neuronal numbers within the graft, the GFP reporter enabled clear distinction of graft derived innervation from the host, Figure 3A–D. Delineation of the striatal volume containing GFP+ fibers revealed a significant increase in grafts in the presence of SDF1 functionalised hydrogels (Cells + SAP-shSDF1: 3.51 + 0.36 mm^3^) compared to other groups (Cells: 2.38 + 0.22; Cells+SAP: 1.51 + 0.40; Cells+sSDF1: 2.51 + 0.38), Figure 3E. Note, in the absence of SDF1, the hydrogel significantly impeded fiber outgrowth and reinnervation of the host tissue.

The capacity for grafted DA neurons to restore motor function is dependent on their capacity to reinnervate the dorsolateral striatum in the rodent brain. However, assessment of this region of the host brain (see asterisk in Figure 3F, indicative of site of sampling) revealed highly variable fiber density, influenced by small graft size and the variable placement of grafts within the striatum (Data not shown). Consequently, the density of graft innervation was assessed at the lateral edge of the graft-host border, (boxed area, Figure 3F). Reflective of innervation volume assessment, only grafts in the presence of the unfunctionalized hydrogel (Cells + SAP) showed a significant reduction in innervation density at the graft-host border, compared to Cells + SAP-shSDF1, indicative of the gel impeding fiber outgrowth Figure 3G–K. These findings highlight the benefit of sustained SDF1 delivery on promoting reinnervation of the host.

### 2.5. Functionalised Scaffold Increases the Fate Acquisition of A9 DA Neurons, a Subpopulation Critical for Motor Function

Critical for the restoration of motor function following transplantation is not only the capacity for the new DA neurons to synaptically integrate, but more specifically the ability of the A9 DA neurons to form connections—a subpopulation of DA cells that normally reside in the SNpc and form the nigrostriatal pathway, innervating the dorsolateral striatum. We therefore examined the number of A9 and A10 neurons within the grafts, assessing GIRK2+GFP+, CALBINDIN+GFP+ and GIRK2+CALBINDIN+GFP+ cells, Figure 4. Grafts in the presence of the SAP hydrogel showed a 2.6-fold increase in GFP+/GIRK2+ co-localised cells (Cells: 1013 ± 121; Cells + SAP: 2589 ± 461). In contrast SDF1 had no impact on A9 fate, with acute delivery failing to significantly increase the number of GIRK2+ cells within the graft (Cells + sSDF1: 1695 ± 461) compared to Cells alone. Sustained SDF1 delivery showing further significant increase in A9 specification compared to Cells + SAP (Cells + SAP-shSDF1: 3774 ± 608), Figure 4A–C. No change in either GFP+/Calbindin+ or GFP+/GIRK2+/Calbindin+ neurons was observed, Figure 4D.

To assess whether the increase in A9-like GFP+/GIRK2+ cells was merely a consequence of increased GFP+ cells (observed to be significantly increase in both Cells + SAP and Cells + SAP-shSDF1 grafted animals, Figure 2K), A9 and A10 fated cells were expressed as a proportion of GFP+ cells within grafts. Only grafts in the presence of the SAP showed a significant increase in the proportion of GIRK2+ cells (Cells: 53 + 4% GFP+GIRK2+/GFP+; Cells + SAP: 69 + 4%, *p* = 0.046; Cell + SAP-shSDF1: 67 + 2%, *p* = 0.125), suggesting that the laminin-based signalling presented by the SAP hydrogel was capable influencing DA fate acquisition Figure 4D.

## 3. Discussion

The capacity to improve the survival, differentiation and/or integration of dopamine (DA) replacing cell grafts will have a significant impact on the number of cells required for grafting, and the predictability of graft function. While modifying one variable at a time can enable assessment of the impact in a biological context, presenting multiple biometric cues can often have a synergistic effect. Here we sought to mimic key features of the physical and trophic extracellular matrix environment of neural tissue—fabricating a customised tissue-specific scaffold, presenting the functional laminin-binding epitope (IKVAV), known to influence cell adhesion, proliferation, differentiation, and plasticity. Furthermore, we encapsulated SDF1 into the gel to sustain delivery. We showed the synergistic capacity of the SAP-shSDF1 hydrogel to increase graft size, DA neuronal numbers, innervation and most critically the proportion of A9-like DA neurons within a graft.

Since the first preclinical and clinical evidence for successful dopamine cell replacement there have been many efforts to improve transplant outcomes. Vulnerability of the donor cells is now recognised as one of the most critical factors contributing to graft success. When donor cells are isolated from fetal tissue, they are stripped of their anchorage to the ECM, activating cell death cascades. Cells then succumb to the physical shear forces of being injected through a fine needle/capillary, and finally these immature progenitors are implanted into the adult host brain, devoid of age-appropriate architecture or trophic support. Such traumas have highlighted the need for a more supportive environment for the donor cells, however the attention has remained on trophic support, with little consideration for the physical environment. Evidence supporting the physical requirements of the graft are highlighted in series of additive studies showing that: (i) the co-grafting of ensheathing cells or sciatic nerve support DA grafts [41,42,43,44], (ii) newly integrating DA neurons utilise residual fibre bundles to act as scaffolding in axonal pathfinding [45] and that (iii) modulation of adhesion proteins L1 and ALCAM influence DA plasticity [46,47,48].

While biomaterials have been trialled and developed in other neural injuries—often to fill tissue voids and bridge gaps within fibre bundles, they have been less commonly employed to support grafts where the host tissue architecture remains intact. We were the first to demonstrate a benefit for utilising a (xyloglucan) hydrogel to support VM fetal tissue grafts [24], also demonstrated by others using a collagen-based hydrogels [49]. In both instances the biomaterials provided only modest effects, likely underpinned by the hydrogel composition and mode of synthesis, with the greatest benefit on graft outcome observed in their capacity to sustain the delivery of GDNF. A key goal of the present study was to utilise a scaffold that most closely mimicked the host tissue. As nanofibrous structural proteins are a major component of the ECM, and extensive effort has been invested in optimising the many properties of nanofibrous scaffold (such as tuning the stiffness to match the host tissue, as well as understanding the impact of nano- verses microscale morphology on cellular responsiveness), we elected to employ the Fmoc-based SAP system to fabricate a nanofibrous hydrogel, presenting the functional laminin epitope IKVAV. The selection of a laminin-based scaffold stemmed from recent observations for the roles of laminin in the survival and differentiation of DA neurons during VM development [50], as well as our observed improvement in the survival, differentiation and functional integration of stem cell-derived cortical grafts in a model of stroke [36]. The present findings showed the SAP scaffold significantly increased DA neurons within the graft, and positively influencing A9 fate acquisition. Further work is required to understand the specificity of the laminin signalling for the A9 fate change, the impact of other laminin epitopes corresponding to alternate laminin subunits (namely laminin-521, implicated in human dopamine differentiation [51]), and whether distinct laminin-responsive transmembrane integrin proteins may be present on the surface of the subpopulations of DA progenitors.

Recent studies have highlighted the importance of SDF1 in DA development [20,21,22], and the DA differentiation of pluripotent stem cells [16,17,18,19]. Added to this, we previously demonstrated that the co-transplantation of meningeal cells, together with VM fetal grafts, significantly increased the proportion of A9 DA neurons within VM grafts—an effect that was speculated to be modulated via SDF1-CXCR4 signalling [22]. The capacity to directly address this question, however, required the ability control the delivery of SDF1 over a timeframe relevant to the regenerative period, as well as targeted to the graft site, to ensure minimal off-target effects. This targeted temporal and spatial delivery of proteins, drugs and small molecules has, and continues to, be no small feat for regenerative medicine. Extensive efforts have been made to stabilize proteins and drugs for in vivo delivery—including the development of various biomaterials [31]. Here we provided evidence of the instability of SDF1, showing rapid degradation in vitro within minutes, highlighted the need for either sustained infusion in vivo (both a costly and cumbersome exercise) or our elected approach to incorporate the recombinant protein into the hydrogel by shear loading. The SAP hydrogel sustained SDF1 delivery over 28 days, showing a slowing rate of release after 14 days. Further studies, involving immunoblotting against SDF1, will be required to more rigorously assess the duration of SDF1 release from the gels following *in vivo* implantation. None-the-less, the in vitro data highlighting the bulk of the delivery within the first 14 days, corresponds with the timeframe over which fetal tissue grafts differentiate and begin to integrate into the surrounding host. The impact of the sustained SDF1 delivery from the gels could be validated by the significant increase in graft survival as well as DA neurons, and relative proportion of GIRK2+ A9-like DA neurons. The lack of response of the grafted cells to the acute delivery of SDF1, highlights the need to sustain delivery. Further studies (in rats and/ non-human primates for which behavioural testing is notably more reliable than mice) will be required to confirm the functionality of these A9 DA enriched grafts, to reverse motor deficits in the PD brain.

The synergistic benefit of sustaining SDF1 from a laminin-based functional hydrogel was most evident by the increase in graft volume. As no detectable difference in the rare Caspase-3 or KI67+ cells were observed at 12 weeks after grafting between groups, further studies, assessing grafts within the acute period will be required to determine whether the larger grafts were a consequence of cell survival during in vivo delivery and integration, or progenitor proliferation following implantation.

SDF1 has previously been reported to strongly influence the plasticity of DA neurites *in vitro* and within the developing brain [20,21,22]. Here we also demonstrate the positive impact of sustained SDF1 protein delivery on graft plasticity. The level of reinnervation however was surprisingly underwhelming given the almost 3-fold increase in DA neurons in Cells+SAP-shSDF1 grafts. Such observations were likely underpinned by insufficient duration, suboptimal dose, and/or the localisation of delivery to influence plasticity. SDF1 has been shown to attract DA neurites [20,21,22], yet was delivered here within the graft core, thereby failing to provide the necessary gradient to draw fibres into the host tissue. In other efforts to promote plasticity of DA neurons within grafts, trophic cues such as GDNF have been delivered within the target tissue (namely the dorsolateral striatum)—distant to the graft site, where they act to draw growing axons closer [45,52,53]. Similar approaches, to delivery SDF1 within hydrogels distal to the graft core, and within the target tissue, will be required in the future to fully elucidate the benefit of SDF1 in promoting graft plasticity.

These findings align with recent studies also highlighting the benefits of sustained SDF1 delivery from hydrogels to support regeneration in animal models of stroke and traumatic brain injury via mechanism inclusive of promoting differentiation of implanted stromal cells as well as promoting local, host neurogenesis and angiogenesis [54,55].

## 4. Materials and Methods

### 4.1. Self-Assembling Peptide Hydrogel Preparation and Functionalisation

The neural tissue specific self-assembling peptide (SAP) hydrogel, presenting the laminin epitope sequence isoleucine-lysine-valine-alanine-valine (IKVAV), was fabricated using solid phase peptide synthesis, as previously described [33]. Two aspartate amino acid residues were added to the N-terminus of the peptide to lower the pKa. This lower pKa enabled spontaneous self-assembling of the peptides to occur under physiological conditions (pH 7.4). The resultant final peptide had the sequence DDIKVAV. Importantly, by using this method of fabrication, the amino acid sidechains of the peptide moieties were presented on the outer edge of the nanofibres, to create a surface rich in bioactive molecules. 

Gelation was initiated using a pH switch to yield a 20 mg/mL stock [37]. Rheology was performed to verify that the modulus of the gel (stiffness) was similar to the mouse brain and thereby amenable to supporting the implanted cells. In addition, fourier transform infrared (FTIR), circular dichroism (CD) spectroscopy, and transmission electron microscopy (TEM) were conducted to confirm appropriate structure of the fabricated nanofibrils [33].

Application of shear force, through titration or vortexing, reversed the gel to an aqueous state and was necessary for the entrapment of SDF1 protein for subsequent *in vivo* delivery. This shear-loading of the protein during the aqueous phase of the hydrogel enabled homogeneous distribution throughout the gel. To assess the release kinetics of the protein from the gel, SDF1 (100 ng, R&D Systems, Minneapolis, MN, USA) was loaded into the SAP (100 µL) by reverse shearing to an aqueous state and then cast into wells of a 96-well plate. PBS (200 µL) was added to the well and incubated at 37 °C. The supernatant was collected at pre-determined intervals over 28 day period. All supernatant samples were stored at −80 °C until the time of analysis. The SDF1 levels were quantified by enzyme linked immunosorbent assay (ELISA), using previously described methods [23,24]. Release profiles were performed in triplicate wells for each time point and repeated on 3 independent experiments.

### 4.2. Ventral Midbrain Foetal Tissue

All animal procedures were conducted in agreement with the Australian National Health and Medical Research Council’s published Code of Practice for the Use of Animals in Research, and experiments approved by The Florey Institute of Neuroscience and Mental Health Animal Ethics committee (#18-005, Approved February 2018). With VM DA neurogenesis, peak CXCR4 expression and in vitro responsiveness of VM DA progenitors to SDF1 protein occurring between embryonic day 10.5–12.5 (E10.5-E12.5) in mice [20,21,22], we nominally elected to utilize E11.5 VM progenitors as the donor cell for transplantation. 

Mice expressing green fluorescent protein driven by the tyrosine hydroxylase promoter (TH-GFP, [56]) were used for time mating—noting TH as the rate-limiting enzyme in DA synthesis and common marker of DA neurons. Consequently, tracking of GFP cells enabled clear identification of graft-derived (GFP+) from residual host (GFP-) DA neurons. E11.5 VM progenitors were harvested from TH-GFP+ embryos using our previously described methods [57]. VM cells were resuspended at 200,000 cells/uL in magnesium and calcium-free Hank’s buffered salt solution containing 0.1% DNase. Cells were stored on ice for the duration of the surgical procedures. At the time of in vivo delivery, the cells were either diluted 1:1 with (i) the SAP hydrogel, following shear reversal to an aqueous state (resulting in a gel concentration of 10mg/ml and cell density of 100,000/uL), or (ii) HBSS media.

### 4.3. Surgical Procedures

Animals were group housed in individually ventilated cages on a 12:12-hour light/dark cycle with ad libitum access to food and water. Surgeries were performed on 28 Swiss mice under 2% isofluorane anaesthesia. A single unilateral injection of the catecholamine-selective neurotoxin 6-hydroxydopamine (6OHDA, 3.2 µg dissolved in 0.02% ascorbic acid in sterile saline) was delivered into the substantia nigra at the following coordinates: 3.2mm posterior, and 1.2mm lateral to bregma and 4.5mm below the surface of the dura, as previously described [58]. Three weeks after lesioning, animals received transplants of one of the following 4 treatments: (i) Cells, (ii) Cells + acute delivery of soluble recombinant SDF1 protein (100 ng, Cells+sSDF1), (iii) Cells + SAP or, (iv) Cells + SAP containing shear encapsulated SDF1 (Cells + SAP-shSDF1) at the following coordinates: 1.0 mm anterior and 2.0 mm lateral to bregma, and 3.0 mm below the dura surface. All animals received a total of 100,000 cells and a final injection volume into the striatum of 1 µL, Figure 1A.

### 4.4. Tissue Processing and Histochemistry

At 12 weeks post-transplantation all animals received an overdose of sodium pentobarbitone (100 mg/kg) and were transcardially perfused with Tyrode buffer followed by 4% paraformaldehyde. The brains were post-fixed for 30 min. and coronally sectioned on a freezing microtome (40 um thickness, 1:12 series) in preparation for histochemical analysis. Primary antibodies and dilution factors were as follows: mouse anti-Calbindin (1:1,500: Swant), rabbit anti-cleaved caspase-3 (Asp175, 1:100; Cell Signalling Technology, Danvers, MA, USA), rabbit anti-GFAP (1:200, DAKO, Santa Clara, CA, USA), mouse anti-CD11b (1:100, Serotec, Oxford, UK), rabbit anti- G-protein-gated inwardly rectifying K+ channel subunit 2 (GIRK2, 1:500; Chemicon, Temecula, CA, USA), chicken anti-GFP (1:1,000; Abcam, Cambridge UK), rabbit anti-GFP (1:20,000; Abcam, Cambridge UK), rabbit anti-KI67 (1:1000, Novocastra, Sheffiled UK), mouse anti-NeuN (1:100; R&D Systems), rabbit anti-TH (1:500, Pel-Freeze), sheep anti-TH (1:800, Pelfreez, Rogers, AR, USA). Secondary antibodies for (i) direct detection were used at a dilution of 1:200—DyLight 488, 549 or 649 conjugated donkey anti-mouse, anti-chicken or anti-rabbit (Jackson ImmunoResearch, West Grove, PA, USA); and (ii) indirect with streptavidin-biotin amplification—biotin conjugated donkey anti-rabbit (1:500; Jackson ImmunoResearch) followed by peroxidase conjugated streptavidin (Vectastain ABC kit, Vector laboratories, Burlingame, CA, USA), or 649 conjugated streptavidin (1:200; Jackson ImmunoResearch). Total cells within the grafts were visualized with 4’, 6-diamidino-2-phenylindole (DAPI, 1:5000, Sigma-Aldrich). All fluorescent images were captured using a Zeiss Axio Observer.Z1 epifluorescence or Zeiss confocal microscope system. Bright and dark-field images were obtained using a Leica DM6000 upright microscope. 

GFP expression was used to delineate the graft boundaries and estimate graft volume (delineation of the volume containing GFP+ cell bodies (see dashed line Figure 2) as well as volume of innervation (see dashed line in Figure 3), according to Cavalieri’s principle [59]. Quantification of GFP+ (DA neurons) was performed on live images and total NeuN+ cells from captured fluorescent images. Assessment of GIRK2+ and Calbindin+ cells within the grafts were quantified based on co-localisation with GFP+ from acquired confocal images.

The density of graft-derived GFP+ DA fibres emanating from the graft was assessed within the dorsolateral striatum (the predominant target of A9 DA neurons and underpinning motor function) as well as immediately lateral to the graft-host border. The graft-host boundary was classified as the border encapsulating all GFP+ cell bodies (graft) and the (host) striatal tissue containing GFP+ fibers—see Figure 3F. Sampling was performed from 3 sections (spanning 1.44 mm) within the striatum. Images were captured at 40X magnification on a Leica 6000 microscope and the GFP fibers were isolated on colour inverted images using the colour range tool on Photoshop (Adobe). Data is expressed as percentage of immunoreactive pixels. All areas were captured in triplicate with conserved settings.

The immunological response of the host to the implantation of the cell grafts (in the presence or absence of the hydrogel and SDF1 protein) was assessed at predetermined sites lateral to the graft-host border (delineated according to GFP labelling), as depicted in Appendix A. The area covered by GFAP or CD11b immunoreactivity was expressed as a percentage of the total pixels in the Cells, Cells+SAP, Cells+sSDF1 and Cells+SAP-SDF1 grafted brains.

### 4.5. Statistical Analysis

All data are presented as mean ± SEM. Statistical tests employed (inclusive of one-way ANOVA and student *t*-tests) are stated in figure legends. Alpha levels of *p* < 0.05 were considered significant, with all statistical analysis performed using GraphPad Prism. * *p* < 0.05, ** *p* < 0.01.

## Figures and Tables

**Figure 1 ijms-23-04646-f001:**
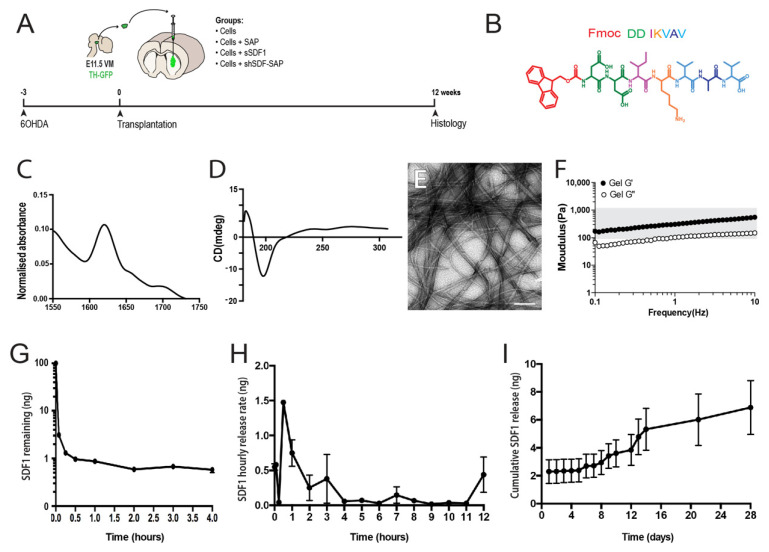
Generation and characterization of SDF1-functionalised hydrogel. (**A**) Schematic of in vivo experimental design highlighting the time frame of 6OHDA lesioning, neural progenitor/biomaterial implantation and histological assessment. (**B**) Illustration of the Fmoc-based peptide sequences, IKVAV. (**C**) Peptide interaction, through hydrogen bonds and secondary β sheet formation was confirmed by Fourier transform infrared (FTIR) of amide I region, showing characteristic peak for β sheets with an observable carbamate peak. (**D**) The stable formation of a dominant assembly, showing transition peaks at 200 nm, was confirmed by circular dichroism spectroscopy. (**E**) Representative electron micrograph illustrating the nanofibril structural organisation. (**F**) Storage (G′, black circles) and loss (G″, white circles) modulus for the IKVAV SAP hydrogel, showing similar stiffness to the rodent brain (indicated by grey shading). (**G**) Soluble SDF1 recombinant protein in media rapidly degraded (>90% degradation within minutes). (**H**,**I**) Shear entrapment of SDF1 in the SAP hydrogel resulted in sustained release, observed over 28 days. Scale bar: 100 nm.

**Figure 2 ijms-23-04646-f002:**
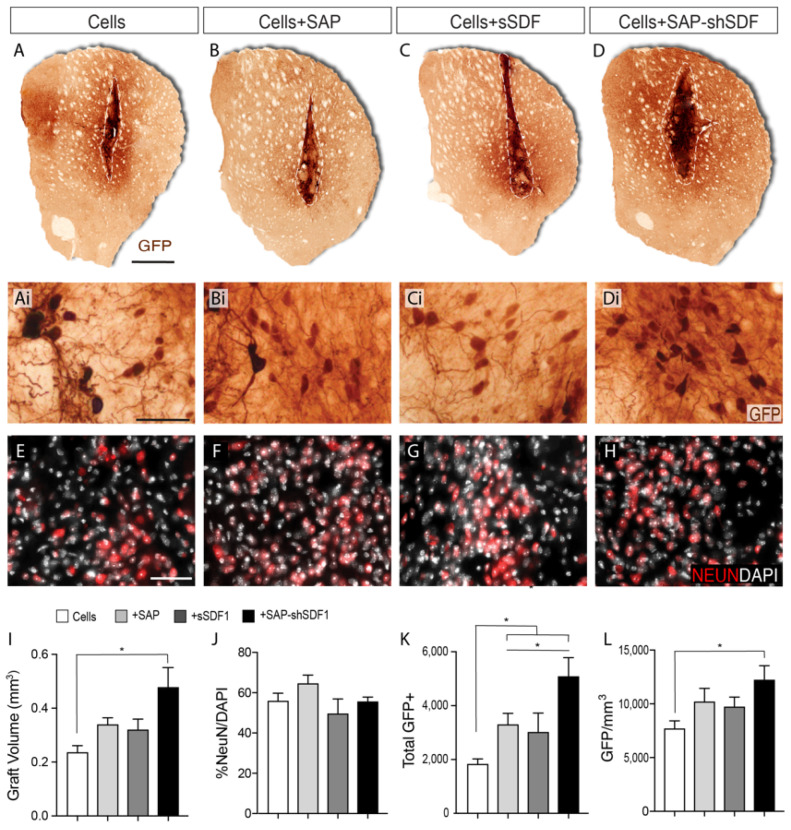
SDF1 functionalised SAP scaffold increase graft size and number of dopamine neurons within foetal tissue neural transplants. (**A**–**D**) Representative photomicrographs of VM foetal tissue grafts showing GFP staining for DA neurons at 12 weeks. Dashed line delineates the graft core, containing the GFP+ DA cell bodies. (**Ai**–**Di**) Higher magnification images from (**A**–**D**) illustrating the density of GFP+ DA neurons and (**E**–**H**) NeuN+ neurons. (**I**) Graft volumetric assessment by GFP labelling revealed that grafts in the presence of the SDF1-functionalised SAP hydrogel were significantly larger compared to the other graft groups. (**J**) NeuN+ neuronal density was unchanged by SDF and/or SAP treatment. (**K**) Quantification of GFP+ cells revealed laminin-based SAP significantly increased DA neuron number, synergistically enhanced by sustained SDF1 delivery. (**L**) GFP+ DA neuron density was significantly increased by the presence of functionalized SAP scaffolds. Data represent Mean ± SEM, *n* = 5–7 Grafts/group. * *p* < 0.05. Scale bar: (**A**–**D**) 1 mm; (**Ai**–**Di**, (**E**–**H**)) 100 μm.

**Figure 3 ijms-23-04646-f003:**
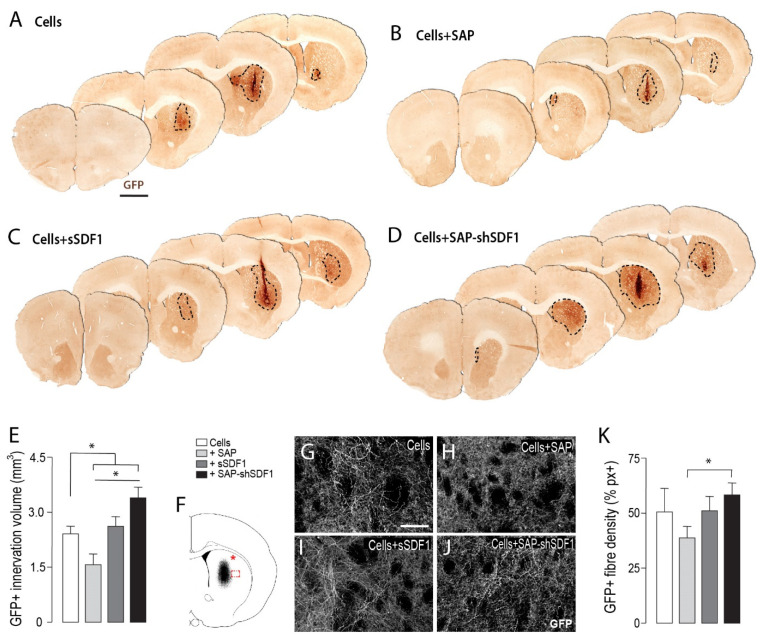
Laminin-based SAP hydrogels, functionalised with SDF1, enhance graft innervation volume and fibre density. (**A**–**D**) Representative photomicrographs of VM foetal tissue grafts in the presence of SAP and/or SDF1, depicting increased GFP+ DA innervation from grafts exposed to SDF1-functionalised scaffolds. Dashed line delineates the GFP+ graft-derived innervation volume. (**E**) Quantification of volumetric GFP+ fibre innervation of the host striatum. (**F**) Schematic illustrating sampling site for assessment of GFP fibre density within the dorsolateral striatum (*) and at the graft-host boundary (boxed area). (**G**–**J**) Representative photomicrographs showing GFP fibre density at the graft host border. (**K**) Quantification of graft-derived GFP fibre density at the graft–host boundary. Data represent Mean ± SEM, *n* = 5–7 Grafts/group. * *p* < 0.05. Scale bars: (**A**–**D**) 2 mm, (**G**–**J**) 200 μm.

**Figure 4 ijms-23-04646-f004:**
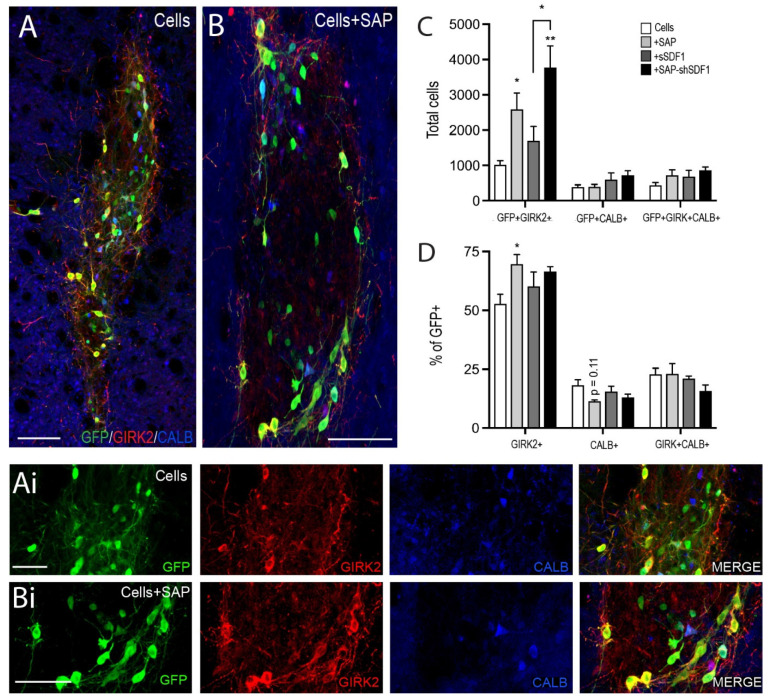
Laminin-based SAP hydrogels bias A9 DA specification. (**A**,**B**) Representative photomicrographs illustrating the subpopulations of DA neurons depicted by GFP+ co-expression with or without GIRK2 and/or Calbinidn (CALB+) within a (**A**) Cells or (**B**) Cells + SAP graft, at 12 weeks. (**Ai**,**Bi**) High power images from (**A**,**B**). (**C**) Quantitative assessment of GFP+ DA neurons co-expressing GIRK2+, CALB+ and GIRK + CALB+, noting SAP scaffolds promoted GIRK2+ DA neuron maturation. (**D**) Expressed as a proportion of GFP+ DA cells, only grafts in the presence of the tissue-specific IKVAV SAP hydrogel showed a significant increase in the percentage of TH+/GIRK2+ (A9-like DA neurons). Data represent Mean ± SEM, *n* = 5–7 Grafts/group. * *p* < 0.05, ** *p* < 0.01, with significant differences compared to the Cells group (unless indicated otherwise). Scale bar: (**A**,**B**) 100 μm; (**Ai**,**Bi**) 50 μm.

## Data Availability

Not applicable.

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
