# Peer review of "Extracellular Matrix Biomimetic Hydrogels, Encapsulated with Stromal Cell-Derived Factor 1, Improve the Composition of Foetal Tissue Grafts in a Rodent Model of Parkinson’s Disease"

_ijms, 2022, doi:10.3390/ijms23094646_

Round 1
Reviewer 1 Report
I commend the authors for the detailed answers to the questions and concerns that were raised. However, the improved manuscript still raises a concern and demand additional revision
Authors should mention and introduce the recommended literature included in the first round of revision based on the previous use of hESC-derived midbrain dopamine neurons, which have shown significant impact in the structural and functional reconstruction of nigra-striatal pathway (https://doi.org/10.1016/j.stem.2020.08.014, https://doi.org/10.1016/j.celrep.2019.08.058, https://doi.org/10.1016/j.stem.2021.01.004). Authors need to explain more convincingly to the readers of IJMS why their study is a breakthrough strategy in the treatment of Parkinson’s disease and the SDF1-hydrogel approach goes beyond actual studies that are now translating into clinical trials without the need of complex designs (for example intracerebral implantation of hydrogels of hard implementation under GLP and subsequent clinical conditions). The authors argue in the need of improvement of graft outcomes with rodent fetal tissue while a significant engraftment of human derived tissues (w/o SDF-1) has been demonstrated in immuno-deficient animals (3 recommended papers). Do the authors think that the in vivo preclinical validation of human products that is usually performed in immunodeficient animals (under GLP) might hide possible defects on hESC-derived midbrain dopamine engraftability in human immunocompetent patients?
Minor:
In discussion “14” (line 454) is repeated twice
Author Response
Comment 1.1: Authors should mention and introduce the recommended literature included in the first round of revision based on the previous use of hESC-derived midbrain dopamine neurons, which have shown significant impact in the structural and functional reconstruction of nigra-striatal pathway https://doi.org/10.1016/j.stem.2020.08.014, https://doi.org/10.1016/j.celrep.2019.08.058, https://doi.org/10.1016/j.stem.2021.01.004).
Response: This is indeed an important point and is acknowledged in the manuscript – citing the exciting progress to human PSC-derived grafting in. preclinical and now clinical trials. There are many, many studies in this space over the past decade (since the first success using bona fide ventral midbrain dopamine progenitors by Lorenz Studers group in 2011 – Kriks et al., 2011, Nature) including several contributions of our own (to name a few - Gantner et al., Cell Stem Cell, 2020; Moriarty et al., Cell Stem Cell 2022, de Luzy et al, Nature Comms, 2019; de Luzy et al, Journal of Neurosci, 2019). We therefore elected to cite a review that we believe encapsulates this progress (see review – Barker et al., 2017). Referred to in lines 60-61.
Comment 1.2: Authors need to explain more convincingly to the readers of IJMS why their study is a breakthrough strategy in the treatment of Parkinson’s disease and the SDF1-hydrogel approach goes beyond actual studies that are now translating into clinical trials without the need of complex designs (for example intracerebral implantation of hydrogels of hard implementation under GLP and subsequent clinical conditions).
Response: We have revised the introduction to more clearly highlight that while there has been exciting progress with human PSC-grafting now advancing to clinical trial, there is, as with all new therapies, room for improvement. This need for improvement is well acknowledged by the field – with this area of preclinical grafting remaining highly active. We have, for example, addressed strategies to improve the purity of grafts (through cell sorting prior to grafting), adopted approaches to ensure graft safety by incorporation of suicide genes, targeted at eliminating proliferative populations in the grafts and employed combined gene therapy to promote graft plasticity. The present work is another example of needing to refine graft – in this instance, focused on promoting survival and maturation of the grafts via sustained SDF1 delivery. See lines 62-66.
Comment 1.3: The authors argue in the need of improvement of graft outcomes with rodent fetal tissue while a significant engraftment of human derived tissues (w/o SDF-1) has been demonstrated in immuno-deficient animals (3 recommended papers).
Response: See above comment regarding the necessity for ongoing research to improve and refine neural grafting – to build upon the success of human PSC-derived grafts. The use of rodent donor tissue provides a valuable preclinical ‘tool’ to accelerate progress – with grafts requiring just weeks to mature and functionally integrate (compared to months needed for human neural grafts) – see manuscript lines 117-121. This approach enables us to rapidly identify whether an approach (such as sustained SDF1 delivery) is likely to have a positive impact on graft outcomes, before adopting it for human PSC-derived grafts (that, in addition to time, are also notable more expensive due to the employment of immune compromised rodents to perform xenografting, and costs associated with culturing human PSCs). This is a common approach we and others have, and continue to adopt. By way of example in 2013 we demonstrated the benefit of GDNF to promote survival and plasticity of DA grafts using fetal mouse donor tissue (Kauhausen et al, 2013, JPhysiol) . More recently we have shown similar findings when grafting human PSC-derived progenitors in the presence of GDNF (Gantner et al., Cell Stem Cell, 2020; Moriarty et al., Cell Stem Cell 2022).
Comment 1.4: Do the authors think that the in vivo preclinical validation of human products that is usually performed in immunodeficient animals (under GLP) might hide possible defects on hESC-derived midbrain dopamine engraftability in human immunocompetent patients?
Response: We have no reason to believe that the current hPSC-derived VM progenitors will not engraft in the current clinical trials. Noting that within these trials ALL patients are receiving immune suppression (for at least 12 months). Confidence of success of engraftment is based upon >30years of preclinical and clinical research demonstration good translation of findings from rodent to human. This has included the progression of rodent fetal grafting (in rodents) to human fetal grafting (in immune compromised rodents) to human fetal grafting in clinical trials (in immune compromised patients). More recently (in the past decade) has focused on navigating the progression of human PSC-derived VM DA progenitor grafting (in immune compromised rodents) and now clinical trials (in immune compromised patients).
Minor comment: In discussion “14” (line 454) is repeated twice.
Response: amended accordingly
Reviewer 2 Report
The manuscript is improved according to reviewer's recommendations
Author Response
We appreciate the reviewers support of our study and recommendation to accept the manuscript.
Round 2
Reviewer 1 Report
Still I found the justification argued by the authors limited.
The justification should not be based in the previous experience of the authors in the field, as this reviewer is evaluating the current manuscript in the context of current literature and actual trends (not only the authors studies). The authors state that the aim of this manuscript is justified in the existence of a “room for improvement”, but it looks that authors are intentionally ignoring at least three seminal and recent studies where a complex design (biomaterial scaffold) was not needed (in fact one of these studies defines a preclinical validation under GLP; currently on clinical trial and non-mentioned by the authors).
For unknown reasons, authors decline to introduce in the manuscript precisely these recent papers (years 2020/2021) in the field published in top journals (this reviewer is not author/coauthor or have conflict of interest with regards to these published papers), avoiding the readers of IJMS to have a global perspective of this topic in relation with the "room improvement" stated by the authors.
Author Response
Reviewer comment 1.1: The justification should not be based in the previous experience of the authors in the field, as this reviewer is evaluating the current manuscript in the context of current literature and actual trends (not only the authors studies). The authors state that the aim of this manuscript is justified in the existence of a “room for improvement”, but it looks that authors are intentionally ignoring at least three seminal and recent studies where a complex design (biomaterial scaffold) was not needed (in fact one of these studies defines a preclinical validation under GLP; currently on clinical trial and non-mentioned by the authors). For unknown reasons, authors decline to introduce in the manuscript precisely these recent papers (years 2020/2021) in the field published in top journals (this reviewer is not author/coauthor or have conflict of interest with regards to these published papers), avoiding the readers of IJMS to have a global perspective of this topic in relation with the "room improvement" stated by the authors.
Author Response: In due respect, we were not ignoring the suggested papers – rather highlighting that there is significantly more literature in this space than these 3 papers. We provided some of our own contributions in this space (advances in human PSC-derived grafts for PD), in the ‘response to reviewer’ document and not the manuscript, as mere examples from a single group). To please the reviewer, however, we have now removed reference to the review and replaced it with the 3 suggested papers. For clinical context, we have also included the first paper (Song et al 2019) that reports on human PSC-grafting in a patient with PD.
We have also included references (original articles and a review) that speaks to the need/supports the statement regarding the need for further improvements within the field – lines 62-66.
This manuscript is a resubmission of an earlier submission. The following is a list of the peer review reports and author responses from that submission.
Round 1
Reviewer 1 Report
In this study, Vanessa Penna and colleagues report a bioengineered scaffold for the delivery of SDF1 to enhance graft survival, maturation of the A9 dopaminergic subpopulation and plasticity in a Parkinson’s mouse model. The study is interesting although the novelty is limited, as SDF1 has been delivered alone or from different scaffolds in the treatment of neural injuries. Authors should address a number of unresolved questions before considering the manuscript for publication
Conceptual framework:
a) The authors propose the use of a hydrogel to deliver SDF1 in order to enhance the engraftment and reinnervation capacity of VM progenitors. Although the experimental approach proposed by the authors is legitimate, the authors ignore previous seminal studies, where the simple transplantation of hESC-derived midbrain dopamine neurons (without SDF1 or hydrogel) produces very convincing evidence of structural and functional nigra-striatal circuit reparation (see for example, https://doi.org/10.1016/j.stem.2020.08.014, https://doi.org/10.1016/j.celrep.2019.08.058, https://doi.org/10.1016/j.stem.2021.01.004). In fact, one of this approaches are currently on going in clinical trial (https://clinicaltrials.gov/ct2/show/NCT04802733). Thus, it seems that there is not need for a more complex design (cells, hydrogel and SDF1), at least, in this specific neurological problem
b) Although I understand that authors pursue to establish a Proof-of-Concept of the in vivo effect of SDF-1 delivered from hydrogels, authors did not perform behavioral testing to infer positive or negative effects linked with this particular approach.
c) The authors offer a simplistic scenario with GFP grafts. Although GFP grafts have been used for others and the same authors in this and previous studies, and in this manuscript GFP cells were used in all groups (control and experimental); have the authors considered that GFP cells will be recognized by microglia and infilitrating inflammatory cells as “strange” cells in a non-GFP host ? See for example https://doi.org/10.1038/sj.gt.3300951
d) I conveniently consider to introdue in the discussion four seminal studies of SDF1 delivered from hydrogels to treat neurological problems (https://www.sciencedirect.com/science/article/pii/S2452199X20301985
;https://doi.org/10.1016/j.biomaterials.2015.05.025
; https://doi.org/10.1038/s41598-019-45238-4; https://doi.org/10.3390/ijms22179609)
Experimental studies:
1) Please specify in all legend figures or the figures which groups were statistically compared. In the figures, asterisks (related with probability of rejecting null hypothesis) appears in the top of several vertical bars but information on this aspect (which groups?) is not provided
2) Authors should measure the in vivo content of SDF1 (time course) after injection (alone or encapsulated inside hydrogels)
3) It is unclear the study performed in figure 1G. To explain the “fragility” and degradability of SDF1, the authors designed an in vitro study that is not physiological, as factors, proteins, molecules and concentrations present in the culture media are very distinct from the brain tissue. Instead of that, authors should measure the in vivo content of SDF1 after injection to justify the encapsulation inside a hydrogel. Although it seems that in several readouts there is a clear effect of SDF1 when was encapsulated in the hydrogel, in other studies it is not, as for example the volume of innervation was not different between SDF1 and SDF1 encapsulated (figure 3E)
4) Considering again the figure 1G, It is not clear why SDF-1 degradation was expressed as a percentage instead of in absolute numbers. Perhaps data in figure 1G, even expressed as percentage, indicates that SDF-1 levels at 12 hours were significantly superior to SDF-1 delivered from hydrogels (that is expressed as real amount)
5) Did the authors examine the influence of possible “prolonged” (not yet examined in vivo) delivery of SDF1 in the recruitment of inflammatory cells (central and peripheral) towards the graft ? The blood-brain-barrier is disrupted in Parkinson´s disease (https://doi.org/10.3389/fphys.2020.593026) or after craniotomies (previous to striatal injection), thus, peripheral inflammatory leukocytes might be attracted by a SDF1 gradient, creating an unfavorable environment for functional repairing
6) Striatal injection: cell concentration is very high (100x106/ml). Did the authors treated with heparin/DNase before injection to prevent cellular aggregates?
7) The authors mention the benefit of sustained SDF1 delivery on promoting reinnervation of the host. However, how the authors distinguish real reinnervation (graft-host border) from just simple proliferation and migration of DA cells from the point of injection towards the border? Did the authors perform specific studies to determine axonal projections? Did the authors found TH-GFP positive cells in substantia Nigra?
8) A9 and A10 mature phenotypes can have their origin in the initial graft (before transplantation). What is the percentage of A9 and A10 phenotypes in embryonic TH-GFP positive cells? is it null?
Minor:
Please be consistent, use SDF-1 or SDF1 (not different abbreviations)
Methods: please give details of sterotaxic coordinates for OHDA-injection and graft implantation
Reviewer 2 Report
Review of a manuscript “Extracellular Matrix Biomimetic Hydrogels, Encapsulated with Stromal Cell-Derived Factor-1, Improves the Composition of Fetal Tissue Grafts in a Rodent Model of Parkinson’s Disease“ by Vanessa Penna and coauthors submitted to IJMS.
Parkinson’s disease is a severe neurodegenerative disorder, for which no efficient medication is yet developed. Cell replacement therapy is an attractive option for dopamine replacement using ventral midbrain dopamine neurons, however this approach is impeded by a low rate survival of the progeny tor/neurons. The authors utilized a neural tissue-specific IKVAV SAP hydrogel, shear-loaded with the chemokine Stromal cell-Derived Factor 1 and tested this approach in an animal model of Parkinson’s disease. This is an important area of research and the results will be interesting for the readership of IJMS.
The following corrections should be made:
Introduction
1 The manuscript begins with the abbreviation of Parkinson’s disease as (PD), however, in the following text the full name is often used, alternation the name as Parkinson’s disease or Parkinson’s Disease
2 After the sentence “Parkinson’s disease (PD) is a progressive neurodegenerative disorder where the loss of ventral midbrain (VM) dopamine (DA) neurons underpins disturbances in motor function” the authors should add a reference to a review on PD ”Emamzadeh FN et al., Parkinson’s disease: Biomarkers, Treatment, and Risk Factors. Frontiers in Neuroscience, Neurodegeneration, 12, 61230, 2018. https://doi.org/10.3389/fnins.2018.00612
Results
3 Figure 1 “Soluble SDF1 recombinant protein in media rapidly degraded (>90% degradation within minutes. The time scale is not appropriate for this figure. If degradation takes just minutes it is unclear why to present the 12 hour scale.
Discussion
4 “The utility of such functional biomimetic scaffolds may significantly improve the functional outcome of cell therapy for neural injuries, including Parkinson’s Disease.” This is very broad statement. The authors should present a brief description how they see the improvement.
Methods
“Two aspartate amino acid residues were added to the N-terminus of the peptide to lower the pKa, ena- bling spontaneous self-assembling to occur under physiological conditions (pH 7.4), and resulted in a final peptide sequence DDIKVAV”
The style of the sentence should be corrected.